# The Maximal Complexity of Quasiperiodic Infinite Words

Ludwig Staiger 

Institut für Informatik, Martin-Luther-Universität Halle-Wittenberg, D-06099 Halle (Saale), Germany; staiger@informatik.uni-halle.de

**Abstract:** A quasiperiod of a finite or infinite string is a word whose occurrences cover every part of the string. An infinite string is referred to as quasiperiodic if it has a quasiperiod. We present a characterisation of the set of infinite strings having a certain word $q$ as quasiperiod via a finite language $P_q$ consisting of prefixes of the quasiperiod $q$. It turns out its star root $\sqrt[*]{P_q}$ is a suffix code having a bounded delay of decipherability. This allows us to calculate the maximal subword (or factor) complexity of quasiperiodic infinite strings having quasiperiod $q$ and further to derive that maximally complex quasiperiodic infinite strings have quasiperiods *aba* or *aabaa*. It is shown that, for every length $l \geq 3$, a word of the form $a^n b a^n$ (or $a^n b b a^n$ if $l$ is even) generates the most complex infinite string having this word as quasiperiod. We give the exact ordering of the lengths $l$ with respect to the achievable complexity among all words of length $l$.

**Keywords:** quasiperiod; formal language; asymptotic growth; polynomial

## 1. Introduction

In his tutorials [1–3] Solomon Marcus dealt with several properties of infinite words. Among them he considered quasiperiodicity and its influence on measures of symmetry like complexity, recurrence or entropy. One topic of interest was their *subword complexity* (or *factor complexity* [4]). Besides the asymptotic behaviour of the factor complexity, also known as their topological entropy ([4], Section 4.2.2) or [5] Marcus was also interested in the behaviour of the complexity function $f(\xi, n)$ assigning to a natural number $n \in \mathbb{N}$ the number of subwords of the infinite word ($\omega$-word) $\xi$. Here he was also concerned with recurrences in $\omega$-words and their influence to subword complexity. A well-known fact established by Grillenberger is that the asymptotic subword complexity (or topological entropy) of an almost periodic (or uniformly recurrent) $\omega$-word can be arbitrarily close (but not equal) to the maximal subword complexity (see [4], Theorem 4.4.4).

The present paper summarises results on the subword complexity of infinite words obtained in [6–8]. We study in detail the structure of the set of infinite words having a certain word $q$ as quasiperiod and how this is connected with the set of finite words with the same quasiperiod. Moreover, we address a question raised in [9] about the maximally achievable subword complexity of a quasiperiodic infinite word.

A first result shows that for every word $q$ there is a value $\lambda_q, 1 \leq \lambda_q < 2$, such that, for every infinite word $\xi$ with quasiperiod $q$, the complexity function $f(\xi, n)$ is bounded by $O(1) \cdot \lambda_q^n$, and this bound is achieved for certain infinite words having quasiperiod $q$. The maximally possible value for $\lambda_q$ is $\lambda_q = t_P \approx 1.324718$, where $t_P$ is the smallest Pisot-Vijayaraghavan number, that is, the unique real root $t_P$ of the cubic polynomial $x^3 - x - 1$.

As a generalisation of the above-mentioned questions [2,9] we estimate, for every length $n \geq 3$, the values $\lambda_n = \max\{\lambda_q : |q| = n\}$, their ordering and the words $q, |q| = n$, for which $\lambda_q = \lambda_n$. It appears that a two letter alphabet is sufficient for achieving the maximal complexity $\lambda_n$.

In order to prove these properties we start with a general investigation of quasiperiodicity of words (as e.g., in [10–12]) and infinite words.

The paper is organised as follows. After introducing some notation we derive in Section 3 a characterisation of quasiperiodic words and $\omega$-words having a certain quasiperiod $q$. Moreover, we use the finite basis sets $P_q$ and its dual $R_q$ ($\mathcal{L}(q)$ and $\mathcal{R}(q)$ in [12]) from which the sets of quasiperiodic words or $\omega$-words having quasiperiod $q$ can be constructed. In Section 4 it is then proved that the star root of $P_q$ is a suffix code having a bounded delay of decipherability and, dually, the star root of $R_q$ is a prefix code.

This much prerequisites allow us, in Section 5, to estimate the number of subwords of the language $Q_q$ of all quasiperiodic words having quasiperiod $q$. It turns out that $c_{q,1} \cdot \lambda_q^n \leq f(Q_q, n) \leq c_{q,2} \cdot \lambda_q^n$ where $f(Q_q, n)$ is the number of subwords of length $n$ of words in $Q_q$ and $1 \leq \lambda_q \leq t_P$ depends on $q$. We construct, for every quasiperiod $q$, a quasiperiodic $\omega$-word $\xi_q$ with quasiperiod $q$ whose subword complexity $f(\xi_q, n)$ is maximal.

The values $\lambda_q$ turn out to be maximal positive roots of polynomials associated with the star root $\sqrt[*]{P_q}$. Section 6 deals with the properties of those polynomials. This allows to compare the roots $\lambda_q$.

The following Sections 7 and 8 deal with the proof of the above mentioned results on the values $\lambda_q$ and $\lambda_n = \max\{\lambda_q : |q| = n\}$. Here we derive also the complete ordering of the values $\lambda_n$.

## 2. Notation and Preliminaries

In this section we introduce the notation used throughout the paper. By $\mathbb{N} = \{0, 1, 2, \ldots\}$ we denote the set of natural numbers. Let $X$ be an alphabet of cardinality $|X| = r \geq 2$, and let throughout the paper $a, b \in X, a \neq b$, be two different letters. By $X^*$ we denote the set of finite words on $X$, including the *empty word e*, and $X^\omega$ is the set of infinite strings ($\omega$-words) over $X$. Subsets of $X^*$ will be referred to as *languages* and subsets of $X^\omega$ as *$\omega$-languages*.

For $w \in X^*$ and $\eta \in X^* \cup X^\omega$ let $w \cdot \eta$ be their *concatenation*. This concatenation product extends in an obvious way to subsets $L \subseteq X^*$ and $B \subseteq X^* \cup X^\omega$. For a language $L$ let $L^* := \bigcup_{i \in \mathbb{N}} L^i$, and by $L^\omega := \{w_1 \cdots w_i \cdots : w_i \in L \setminus \{e\}\}$ we denote the set of infinite strings formed by concatenating words in $L$. The smallest subset of a language $L$ which generates $L^*$ is called its *star root* $\sqrt[*]{L}$ [13]. It holds

$$\sqrt[*]{L} = (L \setminus \{e\}) \setminus (L \setminus \{e\})^2 \cdot L^*.$$

Furthermore $|w|$ is the *length* of the word $w \in X^*$ and $\mathbf{pref}(B)$ is the set of all finite prefixes of the strings in $B \subseteq X^* \cup X^\omega$. We shall abbreviate $w \in \mathbf{pref}(\eta)$ $(\eta \in X^* \cup X^\omega)$ by $w \sqsubseteq \eta$.

We denote by $B/w := \{\eta : w \cdot \eta \in B\}$ the *left derivative* of the set $B \subseteq X^* \cup X^\omega$. As usual, a language $L \subseteq X^*$ is *regular* provided it is accepted by a finite automaton. An equivalent condition is that its set of left derivatives $\{L/w : w \in X^*\}$ is finite.

The sets of infixes of $B$ or $\eta$ are $\mathbf{infix}(B) := \bigcup_{w \in X^*} \mathbf{pref}(B/w)$ and $\mathbf{infix}(\eta) := \bigcup_{w \in X^*} \mathbf{pref}(\{\eta\}/w)$, respectively. In the sequel we assume the reader to be familiar with basic facts of language theory.

We call a word $w \in X^* \setminus \{e\}$ *primitive* if $w = v^n$ implies $n = 1$, that is, $w$ is not the power of a shorter word, and we call $w \in X^* \setminus \{e\}$ *overlap-free* if none of its proper prefixes is a suffix of $w$. The following facts are known (e.g., [14,15]).

**Fact 1.** *Every word $w \in X^* \setminus \{e\}$ has a unique representation $w = v^n$ where $v$ is primitive.*

**Fact 2.** *Let $q, v, w \in X^*, 0 < |v| < |q|$. If $v \cdot q = q \cdot w$ then $v = u \cdot u', q = (u \cdot u')^\kappa \cdot u$ and $w = u' \cdot u$ for some $u, u' \in X^*, u \neq e$, and $\kappa \in \mathbb{N}$. In particular, $q$ is not overlap-free.*

**Fact 3.** *If $w \cdot v = v \cdot w, w, v \in X^*$ then $w, v$ are powers of a common (primitive) word.*

As usual a language $L \subseteq X^*$ is called a *code* provided $w_1 \cdots w_l = v_1 \cdots v_k$ for $w_1, \ldots, w_l, v_1, \ldots, v_k \in L$ implies $l = k$ and $w_i = v_i$. A code $L$ is said to be a *prefix code* (*suffix code*) provided no codeword is a prefix (suffix) of another codeword.

### 3. Quasiperiodicity

*3.1. General Properties*

The notion of quasiperiodicity can be formalised in the following manner. A finite or infinite word $\eta \in X^* \cup X^\omega$ is referred to as *quasiperiodic* with quasiperiod $q \in X^* \setminus \{e\}$ provided that for every $j < |\eta| \in \mathbb{N} \cup \{\infty\}$ there is a prefix $u_j \sqsubseteq \eta$ of length $j - |q| < |u_j| \leq j$ such that $u_j \cdot q \sqsubseteq \eta$, that is, for every $w \sqsubseteq \eta$ the relation $u_{|w|} \sqsubset w \sqsubseteq u_{|w|} \cdot q$ is valid. Informally, $\eta$ has quasiperiod $q$ if every position of $\eta$ occurs within some occurrence of $q$ in $\eta$ [11,12].

Let for $q \in X^* \setminus \{e\}$, $Q_q$ be the set of quasiperiodic words with quasiperiod $q$. Then $\{q\}^* \subseteq Q_q = Q_q^*$ and $Q_q \setminus \{e\} \subseteq X^* \cdot q \cap q \cdot X^*$. In order to describe the set of quasiperiodic strings having a certain quasiperiod $q \in X^* \setminus \{e\}$ the following definition is helpful.

**Definition 1.** *A family $\left(w_i\right)_{i=1}^{\ell}$, $\ell \in \mathbb{N} \cup \{\infty\}$, of words $w_i \in X^* \cdot q$ is referred to as a $q$-chain provided $w_1 = q$, $w_i \sqsubset w_{i+1}$ and $|w_{i+1}| - |w_i| \leq |q|$.*

It holds the following.

**Lemma 1.**

1.  $w \in Q_q \setminus \{e\}$ *if and only if there is a $q$-chain $\left(w_i\right)_{i=1}^{\ell}$ such that $w_\ell = w$.*
2.  *An $\omega$-word $\xi \in X^\omega$ is quasiperiodic with quasiperiod $q$ if and only if there is a $q$-chain $\left(w_i\right)_{i=1}^{\infty}$ such that $w_i \sqsubset \xi$.*

**Proof.** It suffices to show how a family $\left(u_j\right)_{j=0}^{|\eta|-1}$ can be converted to a $q$-chain $\left(w_i\right)_{i=1}^{\ell}$ and vice versa.

Consider $\eta \in X^* \cup X^\omega$ and let $\left(u_j\right)_{j=0}^{|\eta|-1}$ be a family such that $u_j \cdot q \sqsubseteq \eta$ and $j - |q| < |u_j| \leq j$ for $j < |\eta|$.

Define $w_1 := q$ and $w_{i+1} := u_{|w_i|} \cdot q$ as long as $|w_i| < |\eta|$. Then $w_i \sqsubseteq \eta$ and $|w_i| < |w_{i+1}| = |u_{|w_i|} \cdot q| \leq |w_i| + |q|$. Thus $\left(w_i\right)_{i=1}^{\ell}$ is a $q$-chain with $w_i \sqsubseteq \eta$.

Conversely, let $\left(w_i\right)_{i=1}^{\ell}$ be a $q$-chain such that $w_i \sqsubseteq \eta$ and set

$$u_j := \max\nolimits_{\sqsubseteq} \left\{ w' : \exists i (w' \cdot q = w_i \wedge |w'| \leq j) \right\}, \text{ for } j < |\eta|.$$

By definition, $u_j \cdot q \sqsubseteq \eta$ and $|u_j| \leq j$. Assume $|u_j| \leq j - |q|$ and $u_j \cdot q = w_i$. Then $|w_i| \leq j < |\eta|$. Consequently, in the $q$-chain there is a successor $w_{i+1}$, $|w_{i+1}| \leq |w_i| + |q| \leq j + |q|$. Let $w_{i+1} = w'' \cdot q$. Then $u_j \sqsubset w''$ and $|w''| \leq j$ which contradicts the maximality of $u_j$. $\square$

Lemma 1 yields the following consequences.

**Corollary 1.** *Let $u \in \mathbf{pref}(Q_q)$. Then there are words $w, w' \in Q_q$ such that $w \sqsubseteq u \sqsubseteq w'$ and $|u| - |w|, |w'| - |u| \leq |q|$.*

**Corollary 2.** *Let $\xi \in X^\omega$. Then the following are equivalent.*

1.  *$\xi$ is quasiperiodic with quasiperiod $q$.*
2.  *$\mathbf{pref}(\xi) \cap Q_q$ is infinite.*
3.  *$\mathbf{pref}(\xi) \subseteq \mathbf{pref}(Q_q)$.*

*3.2. Finite Generators for Quasiperiodic Words*

In this part we consider the finite languages $P_q$ and $R_q$ ($\mathcal{L}(q)$ and $\mathcal{R}(q)$ in [12]) which generate the set of quasiperiodic words as well as the set of quasiperiodic $\omega$-words having quasiperiod $q$.

We set

$$P_q := \{v : e \sqsubset v \sqsubseteq q \sqsubset v \cdot q\} = \{v : \exists v'(v' \sqsubset q \wedge v \cdot v' = q)\}. \tag{1}$$

Then we have the following properties.

**Proposition 1.**

1.　$q \in P_q$ and $P_q = \{q\}$ if and only if $q$ is overlap-free.
2.　$Q_q = P_q^* \cdot q \cup \{e\} \subseteq P_q^*$
3.　$\mathbf{pref}(Q_q) = \mathbf{pref}(P_q^*) = P_q^* \cdot \mathbf{pref}(q)$

**Proof.** 1. $q \in P_q$ is obvious and and the equivalence follows immediately from the definition of $P_q$.

2. In order to prove $Q_q \subseteq P_q^* \cdot q \cup \{e\}$ we show that $w_i \in P_q^* \cdot q$ for every $q$-chain $(w_i)_{i=1}^{\ell}$. This is certainly true for $w_1 = q$. Now proceed by induction on $i$. Let $w_i = w_i' \cdot q \in P_q^* \cdot q$ and $w_{i+1} = w_{i+1}' \cdot q$. Then $w_i' \cdot v_i = w_{i+1}'$. Now from $w_i \sqsubset w_{i+1}$ we obtain $e \sqsubset v_i \sqsubseteq q \sqsubset v_i \cdot q$, that is, $v_i \in P_q$.

Conversely, let $v_i \in P_q$ and consider $v_1 \cdots v_\ell \cdot q$. Since $q \sqsubseteq v_i \cdot q$ the family $(v_1 \cdots v_j \cdot q)_{j=0}^{\ell}$ is a $q$-chain. This shows $P_q^* \cdot q \cup \{e\} \subseteq Q_q$.

3. is an immediate consequence of 2.　□

Proposition 1 and Corollary 2 imply the following characterisation of $\omega$-words having quasiperiod $q$.

$$\{\xi : \xi \in X^\omega \wedge \xi \text{ has quasiperiod } q\} = P_q^\omega \tag{2}$$

**Proof.** Since $P_q$ is finite, $P_q^\omega = \{\xi : \xi \in X^\omega \wedge \mathbf{pref}(\xi) \subseteq \mathbf{pref}(P_q^*)\}$.　□

A dual generator of $Q_q$ is obtained by the right-to-left duality of reading words using the suffix relation $\leq_s$ instead of the prefix relation $\sqsubseteq$.

$$R_q := \{v : e <_s v \leq_s q <_s q \cdot v\} = \{v : \exists v'(v' <_s q \wedge v' \cdot v = q)\}. \tag{3}$$

Analogously to Proposition 1 we obtain

**Proposition 2.**

1.　$q \in R_q$ and $R_q = \{q\}$ if and only if $q$ is overlap-free.
2.　$Q_q = q \cdot R_q^* \cup \{e\} \subseteq R_q^*$, and
3.　$\mathbf{pref}(Q_q) = \mathbf{pref}(q) \cup q \cdot \mathbf{pref}(R_q^*)$.

The proof of Items 1 and 2 is similar to the proof of Proposition 1 using the reversed version of $q$-chain, and Item 3 then follows from Item 2. A slight difference appears with an analogy to Equation (2).

$$\{\xi : \xi \in X^\omega \wedge \xi \text{ has quasiperiod } q\} = q \cdot R_q^\omega \subseteq R_q^\omega \tag{4}$$

Here the last inclusion might be proper, e.g., for $q = aba$ where $R_{aba}^\omega = \{ba, aba\}^\omega \neq aba \cdot R_{aba}^\omega$.

An alternative derivation of the languages $P_q$ and $R_q$ can be found in Definition 2 of [12]. Here the borders, that is, prefixes which are simultaneously suffixes of the quasiperiod $q$, are used:

$$
\begin{aligned}
P_q &= \{v : \exists w (w \sqsubseteq q \wedge w <_s q \wedge q = v \cdot w)\}\text{, and}\\
R_q &= \{v : \exists w (w \sqsubseteq q \wedge w <_s q \wedge q = w \cdot v)\}\,.
\end{aligned}
$$

In the subsequent sections we focus on the investigation of $P_q$ due to the left-to-right direction of $\omega$-words.

### 3.3. Combinatorial Properties of $P_q$

We investigate basic properties of $P_q$ using simple facts from combinatorics on words (see e.g., [14–16]).

**Proposition 3.** *$v \in P_q$ if and only if $|v| \le |q|$ and there is a prefix $\bar{v} \sqsubseteq v$ such that $q = v^k \cdot \bar{v}$ for $k = \lfloor |q|/|v| \rfloor$.*

This is an immediate consequence of Fact 2.

**Corollary 3.** *$v \in P_q$ if and only if $|v| \le |q|$ and there is a $k' \in \mathbb{N}$ such that $q \sqsubseteq v^{k'}$.*

Now set $q_0 := \min_{\sqsubseteq} P_q$. Then in view of Proposition 3 and Corollary 3 we have the following canonical representation.

$$
q = q_0^k \cdot \bar{q} \text{ where } k = \lfloor |q|/|q_0| \rfloor \text{ and } \bar{q} \sqsubseteq q_0\,. \tag{5}
$$

We will refer to $q_0$ as the *repeated prefix* and to $k$ as the *repetition factor*. If $|q_0| > |q|/2$, that is, if $k = 1$ we will refer to $q$ as *irreducible*. (Reducible words are also known as periodic words [10,11].)

**Corollary 4.** *Every word $v \in \sqrt[*]{P_q}$ is primitive.*

**Proof.** Assume $v = v_1^l$ for some $v \in \sqrt[*]{P_q}$ and $l > 1$. Then $q \sqsubseteq v^{k'} = v_1^{l \cdot k'}$, and, according to Corollary 3 $v_1 \in P_q$ contradicting $v \in \sqrt[*]{P_q}$.  □

**Proposition 4.** *Let $q \in X^*, q \ne e$, $q_0 = \min_{\sqsubseteq} P_q$, $q = q_0^k \cdot \bar{q}$ and $v \in P_q^* \setminus \{e\}$.*

1.  *If $w \sqsubseteq q$ then $v \cdot w \sqsubseteq q$ or $q \sqsubseteq v \cdot w$.*
2.  *If $w \cdot v \sqsubseteq q$ then $w \in \{q_0\}^*$.*

**Proof.** From Proposition 1.2 we know $v \cdot q \in P_q^* \cdot q \subseteq Q_q \subseteq q \cdot X^*$. Consequently, $q \sqsubseteq v \cdot q$. Then $v \cdot w \sqsubseteq v \cdot q$ implies $v \cdot w \sqsubseteq q$ or $q \sqsubseteq v \cdot w$ according to whether $|w \cdot w| \le |q|$ or not.

Since $q_0 \sqsubseteq v$, it suffices to prove the second assertion for $q_0$. First one observes that, $w \sqsubseteq q$ and $|w| \le |q| - |q_0|$. Thus $w \sqsubseteq q_0^{k-1} \cdot \bar{q}$. Therefore, we have $w \cdot q_0 \sqsubseteq q$ and $q_0 \cdot w \sqsubseteq q$ which implies $w \cdot q_0 = q_0 \cdot w$ and, according to Fact 3, $w$ and $q_0$ are powers of a common word. The assertion follows because $q_0$ is primitive.  □

Next we derive a lower bound on the lengths of words in $P_q \setminus \{q_0\}^*$.
To this end, we use the Theorem of Fine and Wilf.

**Theorem 1** ([17])**.** *Let $v, w \in X^*$. Suppose $v^m$ and $w^n$, for some $m, n \in \mathbb{N}$, have a common prefix of length $|v| + |w| - \gcd(|v|, |w|)$. Then $v$ and $w$ are powers of a common word $u \in X^*$ of length $|u| = \gcd(|v|, |w|)$. (Here $\gcd(k, l)$ denotes the greatest common divisor of two numbers $k, l \in \mathbb{N}$.)*

**Proposition 5.** *Let $q \in X^*, q \ne e$, $q_0 = \min_{\sqsubseteq} P_q$, $q = q_0^k \cdot \bar{q}$ and $v \in P_q \setminus \{q_0\}^*$. Then $|v| > |q| - |q_0| + \gcd(|v|, |q_0|)$.*

**Proof.** If $q_0, v \in P_q$ Corollary 3 and Equation (5) imply that $q$ is a common prefix of $q_0^{k+1}$ and $v^{k'}$ for some $k' \in \mathbb{N}$. If $|v| \leq |q| - |q_0| + \gcd(|v|, |q_0|)$ then by Theorem 1 $q_0$ and $v$ are powers of a common word, that is, $v$ is a power of the primitive word $q_0$. $\square$

**Corollary 5.** $\sqrt[*]{P_q} = P_q \setminus q_0^2 \cdot \{q_0\}^*$

**Proof.** It suffices to show $P_q \cap P_q^2 \cdot P_q^* \subseteq \{q_0\}^*$. To this end observe that in view of Proposition 5 $|v \cdot v'| > |q|$ whenever $v \in P_q \setminus \{q_0\}^*$ or $v' \in P_q \setminus \{q_0\}^*$. $\square$

As an immediate consequence we obtain that $\sqrt[*]{P_q} = P_q$ if and only if $q$ is an irreducible quasiperiod. Moreover, Proposition 5 shows that

$$\sqrt[*]{P_q} \subseteq \{q_0\} \cup \{v' : v' \sqsubseteq q \wedge |v'| > |q| - |q_0| + \gcd(|v'|, |q_0|)\}. \tag{6}$$

*3.4. The Reduced Quasiperiod $\hat{q}$*

Next we investigate the relation between a quasiperiod $q = q_0^k \cdot \bar{q}$ where $q_0 = \min_{\sqsubseteq} P_q$ and $\bar{q} \sqsubset q_0$ and its *reduced quasiperiod* $\hat{q} := q_0 \cdot \bar{q}$. Since $q \in Q_{\hat{q}}$, we have $Q_{\hat{q}} \supseteq Q_q$.

We continue with a relation between $P_q$ and $P_{\hat{q}}$. It is obvious that $q_0^i \in P_q$ for every $i = 1, \ldots, k$ and
$$P_{\hat{q}} \subseteq \{v : \hat{q}_0 \sqsubseteq v \sqsubseteq \hat{q}\}. \tag{7}$$

**Lemma 2** ([7], Lemma 2.2). *Let $q \in X^*, q \neq e, q_0 = \min_{\sqsubseteq} P_q, q = q_0^k \cdot \bar{q}$ and $\hat{q} = q_0 \cdot \bar{q}$ the reduced quasiperiod of $q$. Then*

$$P_q = \{q_0^i : i = 1, \ldots, k-1\} \cup \{q_0^{k-1} \cdot v : v \in P_{\hat{q}}\}.$$

**Proof.** Consider $v \in P_{\hat{q}}$. Then $v \sqsubseteq q_0 \bar{q} \sqsubseteq v \cdot q_0 \bar{q}$, and, consequently, $q_0^{k-1} \cdot v \sqsubseteq q_0^k \cdot \bar{q} \sqsubseteq q_0^{k-1} \cdot v \cdot q_0 \bar{q} \sqsubseteq q_0^{k-1} \cdot v \cdot q_0^k \cdot \bar{q}$, that is, $q_0^{k-1} \cdot v \in P_q$.

Conversely, let $v' \in P_q$ and $v' \notin \{q_0^i : i = 1, \ldots, k-1\}$. Then, according to Proposition 5 there is a unique $v \neq e$ such that $v' = q_0^{k-1} \cdot v$. Now $v' = q_0^{k-1} \cdot v \sqsubseteq q = q_0^k \cdot \bar{q} \sqsubset v' \cdot q = q_0^{k-1} \cdot v \cdot q_0^k \cdot \bar{q}$ implies $v \sqsubseteq q_0 \cdot \bar{q} \sqsubset v \cdot q_0^k \cdot \bar{q}$. Since $|v| \leq |q_0 \cdot \bar{q}|$ and $q_0 \cdot \bar{q} \sqsubseteq q_0^k \cdot \bar{q}$, we have $v \sqsubseteq q_0 \cdot \bar{q} \sqsubset v \cdot q_0 \cdot \bar{q}$. $\square$

Together with Corollary 5 this implies

$$P_q \setminus \{q_0\}^* = \sqrt[*]{P_q} \setminus \{q_0\}^* = q_0^{k-1} \cdot (P_{\hat{q}} \setminus \{q_0\}). \tag{8}$$

Moreover, we have the following.

**Corollary 6.** $|\sqrt[*]{P_q}| = 1$ *if and only if $q \in \{q_0\}^*$ and $q_0$ is overlap-free.*

**Proof.** Since $q_0 \in \sqrt[*]{P_q}$, $|\sqrt[*]{P_q}| = 1$ is equivalent with $\sqrt[*]{P_q} = \{q_0\}$ or, according to Equation (8), with $P_{\hat{q}} = \{q_0\}$. This amounts to $\hat{q} = q_0$ and, following Proposition 1.1 $\hat{q} = q_0$ has to be overlap-free. $\square$

For the repeated prefix $\hat{q}_0$ of $\hat{q}$ we have the obvious relation $|\hat{q}_0| > |\bar{q}|$. In case $\hat{q}_0 \neq q_0$ we can improve this.

**Lemma 3.** *Let $q = q_0^k \cdot \bar{q}$ with $k \geq 2$, $\bar{q} \sqsubset q_0$ and $\hat{q} = q_0 \cdot \bar{q}$. If $\hat{q}_0 \neq q_0$ then*

$$\bar{q} \sqsubset \hat{q}_0 \sqsubset q_0 \text{ and } |\hat{q}_0| > |\bar{q}| + \gcd(|q_0|, |\hat{q}_0|),$$

*and there is a nonempty suffix $v \neq e$ of $q_0$ such that $v \sqsubset \hat{q}_0$ and $v \cdot \bar{q} \sqsubset \hat{q}_0^2$.*

**Proof.** We have $\bar{q} \sqsubseteq q_0$ and, since $q_0 \in P_{\hat{q}}$, also $\hat{q}_0 \sqsubseteq q_0$. Moreover, $\hat{q} \sqsubseteq q_0^2$ and $\hat{q} \sqsubseteq \hat{q}_0^{k'}$ for some $k' \in \mathbb{N}$. Since $q_0 \neq \hat{q}_0$ and both prefixes are primitive words, in view of Theorem 1 as a common

prefix of $q_0^2$ and $\hat{q}_0^{|q_0|}$ the word $\hat{q} = q_0 \cdot \bar{q}$ has to satisfy $|\hat{q}| < |q_0| + |\hat{q}_0| - \gcd(|q_0|, |\hat{q}_0|)$, that is, $|\hat{q}_0| > |\bar{q}| + \gcd(|q_0|, |\hat{q}_0|)$. The assertion $\bar{q} \sqsubset \hat{q}_0 \sqsubset q_0$ now follows from a comparison of the lengths of $\bar{q}, \hat{q}_0 \sqsubseteq q_0$.

Now, let $v$ be the suffix of $q_0$ defined by $\hat{q}_0^{k'} \cdot v = q_0 \sqsubset \hat{q}_0^{k'+1}$. Then $v \sqsubset \hat{q}_0$ and $v \cdot \bar{q} \sqsubset (\hat{q}_0)^2$. □

### 3.5. Primitivity and Superprimitivity

In this section we consider the inclusion relations between the languages $Q_q, q \neq e$. Analogously to the primitivity of words in [10–12] a word was referred to as *superprimitive* if it is not covered by a shorter one. This leads to the following definition.

**Definition 2** (superprimitive). *A non-empty word $q \in X^* \setminus \{e\}$ is superprimitive if and only if $Q_q$ is maximal w.r.t. "$\subseteq$" in the family $\{Q_q : q \in X^* \setminus \{e\}\}$.*

The next proposition relates the irreducibility of quasiperiods to superprimitivity.

**Proposition 6** ([12], Remark 4). *If $q \in X^* \setminus \{e\}$ is superprimitive then $|\min_{\sqsubseteq} P_q| > |q|/2$, and if $|\min_{\sqsubseteq} P_q| > |q|/2$ then $q$ is primitive.*

**Proof.** If $q_0 = \min_{\sqsubseteq} P_q$ and $|q_0| \leq |q|/2$ then $q = q_0^k \cdot \bar{q}$ for some $\bar{q} \sqsubset q_0$. Thus $q \in Q_{q_0\bar{q}}$ and $q_0\bar{q} \notin Q_q$.

As $q = q'^m$ with $m > 1$ implies $|q_0| \leq |q'| \leq |q|/2$, the other assertion follows. □

The converse of Proposition 6 is not valid.

**Example 1.** *Let $q = abaabaababaab$. Then $P_q = \{abaabaab, abaabaababa, q\}$, and $|\min_{\sqsubseteq} P_q| = 8 > 13/2$ but as $abaabaababaab \in Q_{abaab}$ the word $q$ is not superprimitive.*
*The word $q = ababa$ is primitive but $q_0 = ab$ has $|q_0| \leq |q|/2$.*

In contrast to the fact that the word $q_0 = \min_{\sqsubseteq} P_q$ is always primitive, it need not satisfy $|\min_{\sqsubseteq} P_{q_0}| > |q_0|/2$ let alone be superprimitive..

**Example 2.** *$q = aabaaabaaaa$ has $q_0 = aabaaabaa$ which, in turn has $P_{q_0} = \{aaba, aabaaaba, q_0\}$ with $|aaba| = 4 < |q_0|/2$.*

It turns out that every language $Q_v$ is contained in a unique maximal $Q_q$. To this end we derive the following lemma (cf. also [10,11]).

**Lemma 4.** *Let $v \in Q_q$ and $u \in \mathbf{infix}(v) \cap q \cdot X^* \cap X^* \cdot q$. Then $u \in Q_q$.*

For the sake of completeness we give a proof.

**Proof.** We use a maximal $q$-chain $(w_i)_{i=1}^n$ with $w_n = v$. Assume $v = u_1 \cdot u \cdot u_2$. Since $u$ has $q$ as prefix and suffix, there are $1 \leq j \leq l \leq n$ such that $w_j = u_1 \cdot q$ and $w_l = u_1 \cdot u$. Let, for $1 \leq i \leq l - j + 1$, the words $w_i'$ be defined by $w_{i+j-1} = u_1 \cdot w_i'$. Then $(w_i')_{i=1}^{l-j+1}$ is a $q$-chain with $w_{l-j+1} = u$, that is, $u \in Q_q$. □

**Corollary 7.** *If $v \in Q_q \cap Q_u$ and $|q| < |u|$ then $Q_u \subseteq Q_q$.*

The corollary shows that every language $Q_v$ is contained in a unique maximal $Q_q$ and that two languages $Q_u, Q_q$ are either disjoint or compatible w.r.t. set inclusion. The latter is not true for $\omega$-languages.

**Example 3.** *Let $q = aabaa$ and $u = aabaaa$. Then $q^\omega \notin P_u^\omega$, $u^\omega \notin P_q^\omega$ but $P_u^\omega \cap P_q^\omega \supseteq aa \cdot \{baaa, baaaa\}^\omega$.*

## 4. $P_q$ and $R_q$ as Codes

In this section we investigate in more detail the properties of the star root of $P_q$. It turns out that $\sqrt[*]{P_q}$ is a suffix code which, additionally, has a bounded delay of decipherability. This delay is closely related to the largest power of $q_0$ being a prefix of $q$.

According to [14,18–20] a subset $C \subseteq X^*$ is a code of a *delay of decipherability $m \in \mathbb{N}$* if and only if for all $v, v', w_1, \ldots, w_m \in C$ and $u \in C^*$ the relation $v \cdot w_1 \cdots w_m \sqsubseteq v' \cdot u$ implies $v = v'$. Observe that $C \subseteq X^*$ is a prefix code if and only if $C$ has delay 0.

First we show that $\sqrt[*]{P_q}$ is a suffix code. This generalises Proposition 7 of [12].

**Proposition 7.** $\sqrt[*]{P_q}$ *is a suffix code, and* $\sqrt[*]{R_q}$ *is a prefix code.*

**Proof.** Assume $u = w \cdot v$ for some $u, v \in \sqrt[*]{P_q}$, $u \neq v$. Then $u \sqsubseteq q$ and Proposition 4 (2) proves $w \in \{q_0\}^* \setminus \{e\}$. Consequently, $|v| \leq |q| - |q_0|$. Now Proposition 5 implies $v \in \{q_0\}^*$ and hence $u \in \{q_0\}^*$. Since $u, v \in \sqrt[*]{P_q}$, we obtain $u = v = q_0$ contradicting $u \neq v$.

Using the duality of $P_q$ and $R_q$ one shows in an analogous manner that $\sqrt[*]{R_q}$ is a prefix code. □

An easy consequence of Proposition 7 is the Left and Right Normal Form of a quasiperiodic string ([12], Proposition 8).

**Corollary 8** (Normal Form). *Every word $w \in Q_q$ has a unique factorisation $w = v_1 \cdot v_2 \cdots v_n$ into words $v_i \in \sqrt[*]{P_q}$ ($\sqrt[*]{R_q}$, respectively).*

Since $\sqrt[*]{R_q}$ is a prefix code while the words $v \in P_q$ are prefixes of each other, we obtain $|\sqrt[*]{P_q} \cap \sqrt[*]{R_q}| = 1$ generalising Remark 5 of [12]. In fact $\sqrt[*]{P_q} \cap \sqrt[*]{R_q} = \{q\}$ or $\sqrt[*]{P_q} \cap \sqrt[*]{R_q} = \{q_0\}$ depending on whether $q \neq q_0^k$ or not.

We continue this part by investigating the delay of decipherability of $\sqrt[*]{P_q}$. We prove that the delay depends on the repetition factor $k$.

**Theorem 2.** *Let $q \in X^* \setminus \{e\}$, $q_0 = \min_{\sqsubseteq} P_q$, and $|\sqrt[*]{P_q}| > 1$. Then $\sqrt[*]{P_q}$ is a code having a delay of decipherability of $k$ or $k+1$.*

**Proof.** If $|\sqrt[*]{P_q}| > 1$ then in view of Proposition 5 there is a $q' \in \sqrt[*]{P_q}$ with $|q'| > |q| - |q_0|$. Since $q' \in P_q$, we have $q \sqsubseteq q' \cdot q_0 \sqsubseteq q' \cdot q$. Consequently, $q_0 \cdot q_0^{k-1} \sqsubseteq q \sqsubseteq q' \cdot q_0$, that is, the delay of decipherability is at least $k$.

To prove the converse we show that for $q \sqsubseteq q_0^m$ the delay cannot exceed $m$.

Assume the contrary, that is, $v \cdot w_1 \cdots w_{m+1} \sqsubseteq v' \cdot u$ for some words $v, v', w_1, \ldots, w_{m+1} \in \sqrt[*]{P_q}$, $v \neq v'$, and $u \in P_q^*$. From Proposition 4 (1) we obtain $u \sqsubseteq q$ or $q \sqsubseteq u$ and, since $|w_i| \geq |q_0|$, also $q \sqsubseteq w_1 \cdots w_{m+1}$.

If $v \sqsubset v'$, in view of the inequality $|v| + |q| \geq |v'| + |q_0|$ our assumption yields $v' \cdot q_0 \sqsubseteq v \cdot q$. Therefore, $w \cdot q_0 \sqsubseteq q$ for the word $w \neq e$ with $v \cdot w = v'$ and, according to Proposition 4 (2) $w \in \{q_0\}^*$. This contradicts the fact that $\sqrt[*]{P_q}$ is a suffix code.

If $v' \sqsubset v$, then $|u| > |w_1 \cdots w_{m+1}| \geq |q|$, and via $|v'| + |q| \geq |v| + |q_0|$ we obtain $v \cdot q_0 \sqsubseteq v' \cdot q$ from our assumption. This yields the same contradiction as in the case $v \sqsubset v'$.

The observation $q \sqsubseteq q_0^{k+1}$ finishes the proof. □

For $q = q_0^k$ the preceding proof shows the following.

**Corollary 9.** *If $q = q_0^k$ and $|\sqrt[*]{P_q}| > 1$ then $\sqrt[*]{P_q}$ has a delay of decipherability of exactly $k$.*

Thus, if $|\sqrt[*]{P_q}| > 1$ and $q \neq q_0^k$ the code $\sqrt[*]{P_q}$ may have a minimum delay of decipherability of $k$ or $k+1$. We provide examples that both cases are possible.

**Example 4.** *Let $q := aabaaaaba$. Then $q_0 = aabaa$, $k = 1$ and $\sqrt[*]{P_q} = P_q = \{q_0, aabaaaab, q\}$ which is a code having a delay of decipherability 2.*

$$\textit{Indeed}\quad \begin{aligned} aabaaaabaa &= q_0 \cdot q_0 \sqsubseteq q \cdot q_0 \quad \textit{or} \\ aabaaaabaa &= q_0 \cdot q_0 \sqsubseteq aabaaaab \cdot q_0 \,. \end{aligned}$$

Moreover, in Example 4, $q \cdot q_0 \notin Q_q$. Thus our example shows also that $q \cdot P_q^*$ need not be contained in $Q_q$.

**Example 5.** *Let $q := aba$. Then $k = 1$ and $P_q = \{ab, aba\}$ is a code having a delay of decipherability 1.* □

Since $\sqrt[*]{R_q}$ is a prefix code, every $\omega$-word $\xi \in R_q^\omega$ has a unique factorisation into words $w \in \sqrt[*]{R_q}$. For suffix codes the situation is, in general, different. Consider e.g., the suffix code $\{b, ba, aa\}$. Property 4 (ii) of [20] (see also ([21], Proposition 1.9)) shows that codes of bounded delay of decipherability also admit a unique factorisation of $\omega$-words. Thus we obtain from Theorem 2.

**Lemma 5** (Normal Form for quasiperiodic $\omega$-words). *Every $\omega$-word $\xi \in P_q^\omega$ has a unique factorisation $\xi = v_1 \cdot v_2 \cdots v_i \cdots$ into words $v_i \in \sqrt[*]{P_q}$.*

## 5. Subword Complexity

In this section we investigate upper bounds on the the subword complexity function $f(\xi, n)$ for quasiperiodic $\omega$-words. If $\xi \in X^\omega$ is quasiperiodic with quasiperiod $q$ then Proposition 3 and Corollary 3 show $\mathbf{infix}(\xi) \subseteq \mathbf{infix}(P_q^*)$. Thus

$$f(\xi, n) \le |\mathbf{infix}(P_q^*) \cap X^n| \text{ for } \xi \in P_q^\omega \,. \tag{9}$$

Similar to ([22], Proposition 5.5) let $\xi_q := \prod_{v \in P_q^* \setminus \{e\}} v$. This implies $\mathbf{infix}(\xi_q) = \mathbf{infix}(P_q^*)$. Consequently, the tight upper bound on the subword complexity of quasiperiodic $\omega$-words having a certain quasiperiod $q$ is $f_q(n) := f(\xi_q, n) = |\mathbf{infix}(P_q^*) \cap X^n|$. Observe that in view of Propositions 1 and 2 the identity

$$\mathbf{infix}(P_q^*) = \mathbf{infix}(R_q^*) = \mathbf{infix}(Q_q) \tag{10}$$

holds.

The asymptotic upper bound on the subword complexity $f_q(n)$ is obtained from

$$\lambda_q = \limsup_{n \to \infty} \sqrt[n]{|\mathbf{infix}(P_q^*) \cap X^n|}\,, \tag{11}$$

that is, for large $n$, $f_q(n) \le \hat{\lambda}^n$ whenever $\hat{\lambda} > \lambda_q$.

The following facts are known from the theory of formal power series (cf. [23,24]). As $\mathbf{infix}(P_q^*)$ is a regular language the power series $\sum_{n \in \mathbb{N}} f_q(n) \cdot t^n$ is a rational series and, therefore, $f_q$ satisfies a recurrence relation

$$f_q(n + k) = \sum_{i=0}^{k-1} a_i \cdot f_q(n + i)$$

with integer coefficients $a_i \in \mathbb{Z}$. Thus $f_q(n) = \sum_{i=0}^{k'-1} g_i(n) \cdot \theta_i^n$ where $k' \le k$, $\theta_i$ are pairwise distinct roots of the polynomial $t^n - \sum_{i=0}^{k-1} a_i \cdot t^i$ and $g_i$ are polynomials of degree not larger than $k$.

In the subsequent parts we estimate values characterising the exponential growth of the family $\left(|\mathbf{infix}(P_q^*) \cap X^n|\right)_{n \in \mathbb{N}}$. This growth mainly depends on the root of largest modulus among the $\theta_i$ and the corresponding polynomial $g_i$.

First we show that, independently of the quasiperiod $q$, the root $\theta_i$ of largest modulus is always positive and the corresponding polynomial $g_i$ is constant.

In the remainder of this section we use, without explicit reference, known results from the theory of formal power series, in particular about generating functions of languages and codes which can be found in the literature, e.g., in [14,23,24].

### 5.1. The Subword Complexity of a Regular Star Language

The language $P_q^*$ is a regular star-language of special shape. Here we show that, generally, the number of subwords of regular star-languages grows only exponentially without a polynomial factor. We start with some easily derived relations between the number of words in a regular language and the number of its subwords.

**Lemma 6.** *If $L \subseteq X^*$ is a regular language then there is an $m \in \mathbb{N}$ such that*

$$|L \cap X^n| \quad \leq \quad |\mathbf{infix}(L) \cap X^n| \quad \leq \quad m \cdot \sum_{i=0}^{2m} |L \cap X^{n+i}| \tag{12}$$

If the finite automaton accepting $L$ has $m$ states then for every $w \in \mathbf{infix}(L)$ there are words $u, v$ of length $\leq m$ such that $u \cdot w \cdot v \in L$. Thus as a suitable $m$ one may choose the number of states of an automaton accepting the language $L \subseteq X^*$.

A first consequence of Lemma 6 is that the identity

$$\limsup_{n\to\infty} \sqrt[n]{|L \cap X^n|} = \limsup_{n\to\infty} \sqrt[n]{|\mathbf{infix}(L) \cap X^n|} \tag{13}$$

holds for regular languages $L \subseteq X^*$.

In order to derive the announced exponential growth we use Corollary 4 of [25] which shows that for every regular language $L \subseteq X^*$ there are constants $c_1, c_2 > 0$ and a $\lambda \geq 1$ such that

$$c_1 \cdot \lambda^n \leq |\mathbf{pref}(L^*) \cap X^n| \leq c_2 \cdot \lambda^n. \tag{14}$$

A consequence of Lemma 6 is that Equation (14) holds also (with a different constant $c_2$) for $\mathbf{infix}(L^*)$.

### 5.2. The Subword Complexity of $Q_q$

In this part we estimate the value $\lambda_q$ of Equation (11). In view of Equations (10) and (14) the value $\lambda_q$ satisfies the inequality $c_1 \cdot \lambda_q^n \leq |\mathbf{infix}(P_q^*) \cap X^n| \leq c_2 \cdot \lambda_q^n$.

As $P_q^*$ is a regular language Equations (11) and (13) show that

$$\lambda_q = \limsup_{n\to\infty} \sqrt[n]{|P_q^* \cap X^n|}$$

which is the inverse of the convergence radius $\mathrm{rad}\,\mathfrak{s}_q^*$ of the power series $\mathfrak{s}_q^*(t) := \sum_{n\in\mathbb{N}} |P_q^* \cap X^n| \cdot t^n$. The series $\mathfrak{s}_q^*$ is also known as the structure generating function of the language $P_q^*$.

Since $\sqrt[*]{P_q}$ is a code, we have $\mathfrak{s}_q^*(t) = \frac{1}{1-\mathfrak{s}_q(t)}$ where $\mathfrak{s}_q(t) := \sum_{v\in \sqrt[*]{P_q}} t^{|v|}$ is the structure generating function of the finite language $\sqrt[*]{P_q}$. As $\mathfrak{s}_q^*$ has non-negative coefficients Pringsheim's theorem shows that $\mathrm{rad}\,\mathfrak{s}_q^* = \lambda_q^{-1}$ is a singular point of $\mathfrak{s}_q^*$. Thus $\lambda_q^{-1}$ is the smallest root of $1 - \mathfrak{s}_q(t)$. Hence $\lambda_q$ is the largest positive root of the polynomial $p_q(t) := t^{|q|} - \sum_{v\in \sqrt[*]{P_q}} t^{|q|-|v|}$.

**Remark 1.** *If the length of $q_0 = \min_{\sqsubseteq} P_q$ does not divide $|q|$ then $p_q(t)$ is the reversed polynomial of $1 - \mathfrak{s}_q(t)$, that is, has as roots exactly the the inverses of the roots of $1 - \mathfrak{s}_q(t)$.*

*If $|q_0|$ divides $|q|$ then $q \notin \sqrt[*]{P_q}$ (cf. Corollary 5) and $p_q(t)$ has additionally the root $0$ with multiplicity $|q| - |q'|$ where $q'$ is the longest word in $\sqrt[*]{P_q}$.*

Summarising our observations we obtain the following.

**Lemma 7.** *Let $q \in X^* \setminus \{e\}$. Then there are constants $c_{q,1}, c_{q,2} > 0$ such that the structure function of the language $\mathbf{infix}(P_q^*)$ satisfies*

$$c_{q,1} \cdot \lambda_q^n \leq |\mathbf{infix}(P_q^*) \cap X^n| \leq c_{q,2} \cdot \lambda_q^n$$

*where $\lambda_q$ is the largest (positive) root of the polynomial $p_q(t)$.*

**Remark 2.** *One could prove Lemma 7 by showing that, for each polynomial $p_q(t)$, its largest (positive) root has multiplicity 1. Referring to Corollary 4 of [25] (see Equation (14)) we avoided these more detailed considerations of a particular class of polynomials.*

Now we are able to formulate our main theorem.

As quasiperiods $q$, $|q| \leq 2$, have trivially $P_q^* = \{q_0\}^*$, that is, $\lambda_q = 1$, in the sequel we confine our considerations to quasiperiods $q$ of length $|q| \geq 3$, and we will always assume that the first letter of a quasiperiod $q$ is $a \in X$.

Define $Q_{\max} := \{a^n b a^n : n \geq 1\} \cup \{a^n w a^n : |w| = 2, w \neq aa, n \geq 1\}$.

**Theorem 3** (Main theorem). *Let $q \in a \cdot X^*, |q| \geq 3, q \notin Q_{\max}$, be a quasiperiod and $n = \lfloor \frac{|q|-1}{2} \rfloor$. Then $\lambda_q < \lambda_{a^n b a^n}$ or $\lambda_q < \lambda_{a^n b b a^n}$ according to whether $|q|$ is odd or even.*
　　　*Moreover, $\lambda_w < \lambda_{aba} = \lambda_{aabaa}$ if $w \in a \cdot X^* \setminus \{aba, aabaa\}$.*

## 6. Polynomials

Before proceeding to the proof of our main theorem we derive some properties of polynomials of the form $p(t) = t^n - \sum_{i \in M} t^i$, where $M \subseteq \{i : i \in \mathbb{N} \wedge i < n\}$. This class of polynomials includes the polynomials $p_q(t)$ whose maximal roots $\lambda_q$ characterise the growth of $\mathbf{infix}(P_q^*)$ as described in Lemma 7. We focus in results which are useful for comparing their maximal roots.

The polynomials $p(t) \in \hat{\mathcal{P}} := \{t^n - \sum_{i \in M} t^i : \varnothing \neq M \subseteq \{0, \ldots, n-1\}\}$ have the following easily verified properties.

$$p(0) \leq 0, p(1) \leq 0, p(2) \geq 1 \text{ and } p(t) < 0 \text{ for } 0 < t < 1. \tag{15}$$

$$\text{If } \varepsilon > 0 \text{ and } p(t') \geq 0 \text{ for some } t' > 0 \text{ then } p\big((1 + \varepsilon) \cdot t'\big) > 0. \tag{16}$$

Since $p(1) \leq 0$ and $p(2) \geq 1$ for $p(t) \in \hat{\mathcal{P}}$, Equation (16) shows that once $p(t') \geq 0$, $t' \geq 1$, the polynomial $p(t)$ has no further root in the interval $(t', \infty)$ and $p(t) \in \hat{\mathcal{P}}$ has exactly one root in the interval $[1, 2)$. This yields the following fundamental property.

**Property 1.** *If $t_0$ is the positive root of the polynomial $p(t) \in \hat{\mathcal{P}}$ in $[1, 2)$ and $1 \leq t' < 2$ then $p(t') \leq 0$ if and only if $t' \leq t_0$.*

For the roots of maximal modulus we have the following theorem.

**Theorem 4** (Cauchy). *Let $p(t) = \sum_{i=0}^n a_i \cdot t^i$ be a complex polynomial. Then every root $t'$ of $p(t)$ satisfies $|t'| \leq t_0$ where $t_0$ is the maximal root of the polynomial $|a_n| \cdot t^n - \sum_{i=0}^{n-1} |a_i| \cdot t^i$.*

This implies the following property of polynomials $p(t) \in \hat{\mathcal{P}}$.

$$\text{If } p(t) = 0 \text{ then } |t| \leq t_0. \tag{17}$$

From Property 1 we derive the following criterion to compare the maximal roots of polynomials in $\hat{\mathcal{P}}$.

**Criterion 1.** *Let $p_1(t), p_2(t) \in \hat{\mathcal{P}}$ have maximal roots $t_1$ and $t_2$, respectively. Then $p_2(t_1) > 0$ if and only if $t_1 > t_2$.*

We conclude this section with a bound on the maximal root of certain polynomials in $\hat{\mathcal{P}}$.

**Lemma 8.** *Let $p(t) = t^n - \sum_{i=0}^{m} t^i, n > m \geq 1$. Then $p(t) < 0$ for $1 \leq t \leq \sqrt[2n-m]{(m+1)^2}$ and $p(t) > 0$ for $\sqrt[n-m]{m+1} \leq t$.*

**Proof.** The assertion follows from the inequality $t^n - (m+1) \cdot t^m < p(t) < t^n - (m+1) \cdot t^{m/2}$ when $t > 1$. The part $p(t) < t^n - (m+1) \cdot t^{m/2}$ uses the arithmetic-geometric-means inequality $\sum_{i=0}^{m} t^i > (m+1) \cdot \sqrt[m+1]{\prod_{i=0}^{m} t^i} = (m+1) \cdot t^{m/2}$, and the other part is obvious. $\square$

The following special case is needed below in Lemma 12.

**Corollary 10.** *If $p(t) = t^n - \sum_{i=0}^{n-3} t^i, n \geq 4$, then $p(t) < 0$ for $1 \leq t \leq \sqrt[n+3]{(n-2)^2}$.*

The subsequent sections are devoted to the proof of our main theorem.

## 7. Irreducible Quasiperiods

We start with irreducible quasiperiods.

### 7.1. Extremal Polynomials

The polynomials $p_q(t)$ of irreducible quasiperiods have non-zero coefficients only for $|q|$ and $i < \frac{|q|}{2}$. Therefore we investigate the set

$$\mathcal{P} := \left\{ t^n - \sum_{i \in M} t^i : n \geq 2 \wedge \varnothing \neq M \subseteq \{i : i \leq \tfrac{n-1}{2}\} \right\}.$$

Let $p_n(t) := t^n - \sum_{i=0}^{\lfloor \frac{n-1}{2} \rfloor} t^i \in \mathcal{P}$.

**Property 2.** *Let $p(t) \in \mathcal{P}$ a polynomial of degree $n \geq 3$. Then $p_n(t) \leq p(t)$ for $t \in [1,2]$, and $p_n(t)$ has the largest positive root among all polynomials of degree $n$ in $\mathcal{P}$.*

**Proof.** This follows from $t^n - \sum_{i=0}^{\lfloor \frac{n-1}{2} \rfloor} t^i < p(t)$ for $p(t) \in \mathcal{P} \setminus \{p_n(t) : n \geq 3\}$ when $1 < t \leq 2$ and Criterion 1. $\square$

Observe that, for $n \geq 1$,

$$p_{2n+1}(t) = t^{2n+1} - \sum_{i=0}^{n} t^i \text{ and } p_{2n+2}(t) = t^{2n+2} - \sum_{i=0}^{n} t^i.$$

Moreover, the words $a^n b a^n \in Q_{\max}$ and $a^n w a^n \in Q_{\max}, w \in \{xb, bx\}, x \in X$ are the quasiperiods corresponding to the extremal polynomials $p_{2n+1}(t) \in \mathcal{P}$ and $p_{2n+2}(t) \in \mathcal{P}$, respectively.

**Lemma 9.** $Q_{\max} := \{q : q \in a \cdot X^* \wedge |q| \geq 3 \wedge p_q(t) = p_{|q|}(t)\}$

**Proof.** If $q \in Q_{\max}$ then obviously $p_q(t) = p_{|q|}(t)$. Conversely, if $p_q(t) = t^{|q|} - \sum_{v \in \sqrt[*]{P_q}} t^{|q|-|v|} = p_{|q|}(t)$ then $\sqrt[*]{P_q} = \{v : v \sqsubseteq q \wedge |v| > \frac{|q|}{2}\}$. Then, in view of $q \sqsubseteq v \cdot q$, every prefix $w \sqsubseteq q$ of length $|w| < \frac{|q|}{2}$ is also a suffix of $q$. This is possible only for $q \in Q_{\max}$ or $q \in \{a\}^*$. $\square$

In the sequel the positive root of $p_n(t)$ is denoted by $\lambda_n$. From Criterion 1 we obtain immediately.

**Property 3.** *Let $t \geq 1$. We have $t < \lambda_n$ if and only if $p_n(t) < 0$.*

Then Property 2 implies the following.

**Theorem 5.** *If $q \in a \cdot X^*, |q| \geq 3$, is an irreducible quasiperiod then $\lambda_q \leq \lambda_{|q|}$, and $\lambda_q = \lambda_{|q|}$ if and only if $q \in Q_{max}$.*

*7.2. The Ordering of the Maximal Roots $\lambda_n$*

Before we proceed to the case of reducible quasiperiods we determine the ordering of the maximal roots $\lambda_n$. This will not only be interesting for itself but also useful for proving $\lambda_q < \lambda_{|q|}$ when $q$ is reducible (see Equation (28) below).

The extremal polynomials $p_n(t), n \geq 2$, satisfy the following general relations (By convention, $\sum_{i=k}^{m} a_i = 0$ if $k > m$).

$$t \cdot p_{2n}(t) - 1 = p_{2n+1}(t), \tag{18}$$
$$p_{2n+2}(t) - t^2 \cdot p_{2n}(t) = t^{n+1} - t - 1, \tag{19}$$
$$t^{n-2} \cdot p_{2n+1}(t) - (t^n + 1) \cdot p_{2n-1}(t) = \sum_{i=0}^{n-3} t^i, \text{ and} \tag{20}$$
$$t^{n-2} \cdot p_{2n+3}(t) - (t^{n+1} + 1) \cdot p_{2n}(t) = -t^n + \sum_{i=0}^{n-3} t^i. \tag{21}$$

**Lemma 10.** *The polynomials $t^3 - t - 1$ and $t^5 - t^2 - t - 1 = (t^2 + 1) \cdot (t^3 - t - 1)$ have largest positive roots $\lambda_3 = \lambda_5$ among all polynomials in $\mathcal{P}$, $\lambda_5 > \lambda_4$ and $\lambda_{2n-1} > \lambda_{2n+1} > \lambda_{2n}$ for $n \geq 3$.*

**Proof.** From Equation (18) we have $p_{2n+1}(\lambda_{2n}) = -1 < 0$ and, therefore, $\lambda_{2n} < \lambda_{2n+1}$ when $n \geq 1$.

Similarly, Equation (20) yields $p_{2n+1}(\lambda_{2n-1}) = \lambda_{2n-1}^{-(n-2)} \cdot \sum_{i=0}^{n-3} \lambda_{2n-1}^i > 0$ which implies $\lambda_{2n+1} < \lambda_{2n-1}$ for $n \geq 3$ and $\lambda_3 = \lambda_5$ when $n = 2$. □

The largest (positive) root $\lambda_3$ of the polynomial $t^3 - t - 1$ is also known as the smallest Pisot-Vijayaraghavan number.

So far we have ordered the 'odd' roots: $\lambda_3 = \lambda_5 > \lambda_7 > \lambda_9 > \cdots$. Next we are going to investigate the ordering of the 'even' roots $\lambda_{2n}$, $n \geq 2$.

To this end we derive the following bounds.

**Lemma 11.**

1. $\sqrt[3n+1]{n^2} \leq \lambda_{2n} \leq \sqrt[n+1]{n}$ and $\sqrt[3n-1]{n^2} \leq \lambda_{2n-1} \leq \sqrt[n]{n}$ for $n \geq 2$.
2. Let $n \geq 5$. Then $\lambda_{2n} \geq \sqrt[n-1]{2}$.

**Proof.** 1. follows from Lemma 8.

2. We calculate $p_{2n}(\sqrt[n-1]{2}) = 4 \cdot \sqrt[n-1]{4} - \sum_{i=0}^{n-1} \sqrt[n-1]{2^i} \leq 4 \cdot \sqrt[4]{4} - (2 + (n-1)) = 4 \cdot \sqrt{2} - (n+1) < 0$ if $n \geq 5$ and the assertion follows with Property 1. □

**Remark 3.** The lower bound of Lemma 11.2 does not exceed the lower bound in Lemma 11.1. However, the latter is more convenient for the purposes of Lemma 12.

**Lemma 12.** *If $n \geq 5$ then $\lambda_{2n-2} > \lambda_{2n}$ and $\lambda_{2n} > \lambda_{2n+3}$.*

**Proof.** If $t \geq \sqrt[n-1]{2}$ then $t^n - t - 1 \geq t - 1 > 0$. Consequently, Equation (19) and Lemma 11.2 imply $p_{2n-2}(\lambda_{2n}) < 0$ whence $\lambda_{2n} < \lambda_{2n-2}$.

If $n \geq 5$ we have $\sqrt[n+1]{n} \leq \sqrt[n+3]{(n-2)^2}$ and, following Lemma 11.1 $\lambda_{2n} \leq \sqrt[n+3]{(n-2)^2}$. Then Equation (21) yields $-\lambda_{2n} \cdot p_{2n+3}(\lambda_{2n}) = \lambda_{2n}^n - \sum_{i=0}^{n-3} \lambda_{2n}^i$, and Corollary 10 shows $p_{2n+3}(\lambda_{2n}) > 0$ whence $\lambda_{2n} > \lambda_{2n+3}$. □

Since $p_8(\sqrt[3]{2}) > 0$, the proof of Lemma 12 cannot be applied to lower values of $n$. Thus it remains to establish the order of the $\lambda_i$ for $i \leq 13$. To this end, we consider some special identities and use Criterion 3 and Lemma 12.

$$
\begin{aligned}
p_{12}(t) - (t^8 + t^5 + t^4 + t^2 + t) \cdot p_4(t) &= t^2 - 1, \text{ and} & (22) \\
p_{13}(t) - t \cdot (t^8 + t^5 + t^4 + t^2 + t) \cdot p_4(t) &= t^3 - t - 1 = p_3(t). & (23)
\end{aligned}
$$

**Lemma 13.** $\lambda_8 > \lambda_{10} > \lambda_{13} > \lambda_4 > \lambda_{12}$

**Proof.** Lemma 12 shows $\lambda_8 > \lambda_{10} > \lambda_{13}$. Equation (22) yields $p_{12}(\lambda_4) = \lambda_4^2 - 1 > 0$ whence $\lambda_4 > \lambda_{12}$, and Equation (23) yields $p_{13}(\lambda_4) = p_3(\lambda_4) < 0$, that is, $\lambda_{13} > \lambda_4$. This shows our assertion. $\square$

For the remaining part we consider the identities

$$
\begin{aligned}
t^2 \cdot p_{11}(t) - (t^5 + 1) \cdot p_8(t) &= -t^4 + t + 1 = -p_4(t), & (24) \\
p_{11}(t) - (t^5 + 1) \cdot p_6(t) &= t^3 \cdot p_4(t), \text{ and} & (25) \\
t \cdot p_9(t) - (t^4 + 1) \cdot p_6(t) &= -t^3 + 1. & (26)
\end{aligned}
$$

**Lemma 14.** $\lambda_9 > \lambda_6 > \lambda_{11} > \lambda_8$

**Proof.** We use Equations (24)–(26). Then $p_{11}(\lambda_8) = -p_4(\lambda_8) < 0$ implies $\lambda_{11} > \lambda_8$, $p_{11}(\lambda_6) = \lambda_6^3 \cdot p_4(\lambda_6) > 0$ implies $\lambda_6 > \lambda_{11}$, and, finally, $\lambda_6 \cdot p_9(\lambda_6) = -\lambda_6^3 + 1 < 0$ implies $\lambda_9 > \lambda_6$. $\square$

Now Lemma 10, 12–14 yield the complete ordering of the values $\lambda_n$.

**Theorem 6.** *Let $\lambda_n, n \geq 3$, be the maximal root of the polynomial $p_n(t)$. Then the overall ordering of the values $\lambda_n$ starts with*

$$
\lambda_3 = \lambda_5 > \lambda_7 > \lambda_9 > \lambda_6 > \lambda_{11} > \lambda_8 > \lambda_{10} > \lambda_{13} > \lambda_4 > \lambda_{12}
$$

*and continues as follows $\lambda_{2n+1} > \lambda_{2n} > \lambda_{2n+3}, n \geq 7$.*

In connection with Proposition 6 and Corollary 7 we obtain that the Pisot-Vijayaraghavan number $\lambda_3 = \lambda_5$ is an overall upper bound on the values $\lambda_q$.

**Corollary 11.** *If $q \in X^*, |q| \geq 3$, then $\lambda_q \leq \lambda_3 = \lambda_5$.*

From Lemma 11.1 we obtain immediately.

**Corollary 12.** *Let $M \subseteq \mathbb{N} \setminus \{0, 1, 2\}$ be infinite. Then $\inf\{\lambda_i : i \in M\} = 1$.*

## 8. Reducible Quasiperiods

Reducible quasiperiods $q$ have a repeated prefix $q_0 = \min_\sqsubseteq P_q$ with $|q_0| \leq |q|/2$ and a repetition factor $k \geq 2$ such that $q = q_0^k \cdot \bar{q}$ where $\bar{q} \sqsubset q_0$. Moreover $|\bar{q}| < |q_0| \leq |q|/2$. Observe that $q_0$ is primitive.

We shall consider three cases depending on the relation between the lengths $n = |q|$, $\ell = |q_0|$, the length of the suffix $|\bar{q}| < |q_0|$ and the repetition factor $k \geq 2$.

IN the first case $|q_0| + |\bar{q}| \leq 2$, in view of $\bar{q} \sqsubset q_0$, we have necessarily $\bar{q} = e$ and $q \in a^* \cup \{ab\}^*, a, b \in X, a \neq b$ and, therefore, $Q_q = \{q_0\}^*$ and $\lambda_q = 1$.

Let now $|q_0| + |\bar{q}| \geq 3$. We divide the remaining cases according to the additional requirement $|q| - 2|q_0| \geq 3$ and its complementary one $|q| - 2|q_0| \leq 2$. In the latter case we have necessarily $k = 2$ and $|\bar{q}| \leq 2$.

*8.1. The Case $|q_0| + |\bar{q}| \geq 3 \wedge |q| - 2|q_0| \geq 3$*

Thus, the preceding consideration shows that we have $|\bar{q}| \geq 3$ (in particular, if $q = q_0^2 \cdot \bar{q}$) or the repetition factor $k \geq 3$. This implies $|q| = 7$ (where $q = (ab)^3 a$) or $|q| \geq 9$. From Equation (6) we have

$$\sqrt[*]{P_q} \subseteq \{q_0\} \cup \{v : v \sqsubseteq q \wedge |v| > |q| - |q_0| + 1\} \tag{27}$$

This implies that for $|q_0| \leq |q|/2$ the polynomials $p_q(t)$ have non-zero coefficients only for $|q| = n$, $|q| - |q_0| = n - \ell$ and $i < |q_0| - 1$, that is, are of the form $p_q(t) = t^n - t^{n-\ell} - \sum_{i \in M_q} t^i$ where $M_q \subseteq \{i : i < \ell - 1\}$. Therefore, in the sequel we consider the positive roots of polynomials in

$$\mathcal{P}_{\text{red}} := \Big\{ t^n - t^{n-\ell} - \sum_{i \in M} t^i : n \geq 1 \wedge \ell \leq \frac{n}{2} \wedge M \subseteq \{i : i < \ell - 1\} \Big\}$$

Let $p_{n,\ell}(t) := t^n - t^{n-\ell} - \sum_{i=0}^{\ell-2} t^i \in \mathcal{P}_{\text{red}}$ and $\lambda_{n,\ell}$ be its maximal root. (In the preceding paper [8] we used a slightly different definition of $\mathcal{P}_{\text{red}}$, and, therefore, of $p_{n,\ell}(t)$ and $\lambda_{n,\ell}$.) Similar to Property 2, Criterion 3 and Theorem 5 we have the following.

**Property 4.** *Let $n \geq 3, \ell \leq \frac{n}{2}$ and $p(t) \in \mathcal{P}_{\text{red}}$. Then $p(t) \geq p_{n,\ell}(t)$ for $t \in [1,2]$, and $p_{n,\ell}(t)$ has the largest positive root among all polynomials of degree n and parameter $\ell$ in $\mathcal{P}_{\text{red}}$.*

**Lemma 15.** *If $q, |q| = n$, is a quasiperiod with $|q_0| = \ell \leq n/2$ then $p_q(t) \geq p_{n,\ell}(t)$ for $t \geq 1$, in particular, $\lambda_q \leq \lambda_{n,\ell}$.*

**Remark 4.** *In contrast to Property 2 not for every polynomial $p_{n,\ell}(t)$ there is a quasiperiod q such that $p_{n,\ell}(t) = p_q(t)$, see Remark 5 below.*

We have the following relation between the polynomials $p_n(t)$ and $p_{n,\ell}(t)$.

$$p_n(t) - t^\ell \cdot p_{n-2\ell}(t) = p_{n,\ell}(t) - t^{\ell-1}, \text{ for } n - 2\ell \geq 3 \tag{28}$$

This yields

**Corollary 13.** *Let $n - 2 \cdot \ell \geq 3$. If $\lambda_n < \lambda_{n-2\ell}$ then $\lambda_{n,\ell} < \lambda_n$.*

**Proof.** If $\lambda_n < \lambda_{n-2\ell}$ then $p_{n-2\ell}(\lambda_n) < p_{n-2\ell}(\lambda_{n-2\ell}) = 0$. Thus $p_{n,\ell}(\lambda_n) = -\lambda_n^\ell \cdot p_{n-2\ell}(\lambda_n) + \lambda_n^{\ell-1} > 0$, that is, $\lambda_n > \lambda_{n,\ell}$. $\square$

Next we show the relation $\lambda_q < \lambda_{|q|}$ for all quasiperiods $q$ having $|q_0| \leq |q|/2$ and $|q_0| + |\bar{q}| \geq 3$.

**Lemma 16.** *Let $|q| - 2|q_0| \geq 3$ and $|q_0| + |\bar{q}| \geq 3$. Then $\lambda_q < \lambda_{|q|}$.*

**Proof.** Above we have shown that $|q| - 2|q_0| \geq 3$ and $|q_0| + |\bar{q}| \geq 3$ imply $|q| \geq 7$ or $|q| \geq 10$ according to whether $|q|$ is odd or even.

The ordering of Theorem 6 and Corollary 13 show $\lambda_n > \lambda_{n,\ell}$ for all odd values $n \geq 7$ and for all even values $n \geq 12$.

It remains to consider the exceptional case when $n = |q| = 10$. Here $|q| - 2|q_0| \geq 3$ and $|q_0| + |\bar{q}| \geq 3$ imply $\ell = |q_0| = 3$. Consider $p_{10,3}(t) = t^{10} - t^7 - t - 1 = p_{10}(t) - t^2 \cdot p_5(t)$.

From $\lambda_5 > \lambda_{10}$ and $p_{10}(\lambda_{10}) = 0$ we have $p_{10,3}(\lambda_{10}) = -\lambda_{10}^2 \cdot p_5(\lambda_{10}) > 0$, that is, $\lambda_{10,3} < \lambda_{10}$. $\square$

**Remark 5.** *Equation* (6) *shows that for* $n = |q| = 10$ *and* $\ell = |q_0| = 3$ *we have* $\sqrt[*]{P_q} = \{q_0, q\}$, *that is,* $p_q(t) = t^{10} - t^7 - 1$. *Thus there is no quasiperiod* $q$ *such that* $p_q(t) = p_{10,3}(t) = t^{10} - t^7 - t - 1$.

*8.2. The Case* $|q_0| + |\bar{q}| \geq 3 \wedge |q| - 2|q_0| \leq 2$

This amounts to $|q| = 2 \cdot |q_0| + |\bar{q}|$ where $|\bar{q}| \in \{0, 1, 2\}$.

Here we have to go into more detail and to take into consideration also the reduced quasiperiod $\hat{q} = q_0 \cdot \bar{q}$ of $q$ and its repeated prefix $\hat{q}_0 = \min_{\sqsubseteq} P_{\hat{q}}$. Observe that both repeated prefixes $q_0, \hat{q}_0$ are primitive.

For $q = q_0^k \cdot \bar{q}, k \geq 2$, we have from Equations (7) and (8)

$$p_q(t) \in \left\{ t^{|q|} - t^{|q|-|q_0|} - \sum_{i \in M} t^i : M \subseteq \{0, \ldots, |\hat{q}| - |\hat{q}_0|\} \right\}.$$

Observe that $|\hat{q}_0| > |\bar{q}|$ (in view of Lemma 3 even $|\hat{q}_0| > |\bar{q}| + 1$ if $\hat{q}_0 \neq q_0$) and thus $|\hat{q}| - |\hat{q}_0| = |q_0| - (|\hat{q}_0| - |\bar{q}|) < |q_0|$.

Let $\mathcal{P}'_{\mathrm{red}} := \left\{ t^n - t^\ell - \sum_{i \in M} t^i : n > \ell > j \wedge M \subseteq \{0, \ldots, \ell - j\} \right\}$ and $p_{n,\ell,j}(t) = t^n - t^\ell - \sum_{i=0}^{\ell - j} t^i$. Here the parameter $j$ corresponds to the value $|\hat{q}_0| - |\bar{q}|$. Then similar to Property 4 and Lemma 15 we have

**Property 5.** *Let* $n, \ell \geq 3, \ell \leq \frac{n}{2}, \ell > j$, *and* $p(t) \in \mathcal{P}'_{\mathrm{red}}$. *Then* $p(t) \geq p_{n,\ell,j}(t)$ *for* $t \in [1, 2]$, *and* $p_{n,\ell,j}(t)$ *has the largest positive root among all polynomials of degree* $n$ *and parameters* $\ell$ *and* $j$ *in* $\mathcal{P}'_{\mathrm{red}}$.

**Lemma 17.** *If* $q, |q| = n$, *is a quasiperiod with* $|q_0| = \ell \leq n/2$ *and* $|\hat{q}_0| - |\bar{q}| \geq j$ *then* $p_q(t) \geq p_{n,\ell,j}(t)$ *for* $t \geq 1$, *in particular,* $\lambda_q \leq \lambda_{n,\ell,j}$.

We consider the cases $|\bar{q}| \in \{0, 1, 2\}$ separately. In the sequel we shall make use of the relation

$$t^3 - t^2 - 1 \leq t^2 - t - 1 < 0 \text{ for } 1 \leq t \leq \lambda_3 = \max\{\lambda_n : n \in \mathbb{N}\}. \tag{29}$$

*8.2.1. The Case* $q = q_0^2 \wedge |\bar{q}| = 0$

As shown above the case $|q_0| \leq 2$ and $|\bar{q}| = 0$ amounts to $\lambda_q = 1$. Thus we may consider only the case when $|q_0| \geq 3$. Here we have the following relation between $p_{2\ell}(t)$ and $p_{2\ell,\ell,3}(t)$.

$$p_{2\ell}(t) - p_{2\ell,\ell,3}(t) = t^{\ell-2}(t^2 - t - 1) \tag{30}$$

**Lemma 18.** *If* $q = q_0^2$ *and* $|q_0| = \ell \geq 3$ *then* $\lambda_q < \lambda_{|q|}$.

**Proof.** First we suppose $|\hat{q}_0| \geq 3$. Then $|\hat{q}_0| - |\bar{q}| \geq 3$, and Property 5 and Lemma 17 yield $p_q(t) \geq p_{2\ell,\ell,3}(t)$ for $t \in [1, 2]$. Now Equations (29) and (30) show $p_q(\lambda_{2\ell}) \geq p_{2\ell,\ell,3}(\lambda_{2\ell}) = -\lambda_{2\ell}^{\ell-2}(\lambda_{2\ell}^2 - \lambda_{2\ell} - 1) > 0$, that is $\lambda_q < \lambda_{2\ell}$.

It remains to consider $1 \leq |\hat{q}_0| \leq 2$. If $\hat{q}_0 \in a^*$ then $q_0 = a^\ell$ which is not primitive. Thus $\hat{q}_0 = ab$ and, since $q_0$ is primitive, $q_0 = (ab)^m a, m \geq 1$ whence $q = q_0^2 = (ab)^m a \cdot (ab)^m a$.

We obtain $\sqrt[*]{P_q} = \{(ab)^m a \cdot (ab)^i : i = 0, \ldots, m\}$ and, consequently, $p_q(t) = t^{4m+2} + \sum_{i=0}^m t^{2i+1}$. Then (Observe again $\sum_{i=k}^m a_i = 0$ if $k > m$).

$$
\begin{aligned}
p_q(t) - p_{4m+2}(t) &= -t^{2m+1} + \sum_{i=0}^m t^{2i} = -t^{2m+1} + t^{2m} + t^{2m-2} + \sum_{i=0}^{m-2} t^{2i} \\
&= -t^{2m-2} \cdot (t^3 - t^2 - 1) + \sum_{i=0}^{m-2} t^{2i},
\end{aligned}
$$

and from Equation (29) we obtain $p_q(\lambda_{4m+2}) \geq -\lambda_{4m+2}^{2m-2}(\lambda_{4m+2}^3 - \lambda_{4m+2}^2 - 1) > 0$. □

8.2.2. The Case $q = q_0^2 \cdot \bar{q} \wedge |\bar{q}| = 1$

Here we have the following relation between $p_{2\ell+1}(t)$ and $p_{2\ell+1,\ell,2}(t)$.

$$p_{2\ell+1}(t) - p_{2\ell+1,\ell,2}(t) = t^{\ell-1}(t^2 - t - 1) \tag{31}$$

**Lemma 19.** *If $q = q_0^2 \cdot a, a \in X$, then $\lambda_q < \lambda_{|q|}$.*

**Proof.** First we suppose $|\hat{q}_0| - |\bar{q}| \geq 2$. Then $\ell = |q_0| \geq |\hat{q}_0| \geq 3$, and Property 5 and Equation (31) yield $p_q(\lambda_{2\ell+1}) \geq p_{2\ell+1,\ell,2}(\lambda_{2\ell+1}) = p_{2\ell+1}(\lambda_{2\ell+1}) - \lambda_{2\ell+1}^{\ell-1}(\lambda_{2\ell+1}^2 - \lambda_{2\ell+1} - 1)$. The assertion $p_q(\lambda_{2\ell+1}) > 0$, that is $\lambda_q < \lambda_{2\ell+1}$ follows from Equation (29).

It remains to consider $|\hat{q}_0| = 2$. By Lemma 3 $\hat{q}_0 = q_0$ implies $|\hat{q}_0| > |\bar{q}| + 1 = 2$. Hence $\hat{q}_0 = q_0 = ab$, $q = ababa$ and $p_q(t) = t^5 - t^3 - 1 = t^2 \cdot p_3(t) + t^2 - 1$. Then $\lambda_{ababa} < \lambda_5$ follows from $\lambda_5 = \lambda_3$ and $p_q(\lambda_5) = \lambda_5^2 - 1 > 0$. □

8.2.3. The Case $q = q_0^2 \cdot \bar{q} \wedge |\bar{q}| = 2$

Here we have the following relation between $p_{2\ell+2}(t)$ and $p_{2\ell+2,\ell,2}(t)$.

$$p_{2\ell+2}(t) - p_{2\ell+2,\ell,2}(t) = t^{\ell-1}(t^3 - t - 1) = t^{\ell-1} \cdot p_3(t) \tag{32}$$

**Lemma 20.** *If $q = q_0^2 \cdot \bar{q}$ with $|\bar{q}| = 2$ then $\lambda_q < \lambda_{|q|}$.*

**Proof.** First we suppose $|\hat{q}_0| \geq 4$. Then Property 5, Equation (32) and $\lambda_{2\ell+2} < \lambda_3$ yield $p_q(\lambda_{2\ell+2}) \geq p_{2\ell+2,\ell,2}(\lambda_{2\ell+2}) = -\lambda_{2\ell+2}^{\ell-1} \cdot p_3(\lambda_{2\ell+2}) > 0$, that is, $\lambda_q < \lambda_{2\ell+2}$.

It remains to consider $|\hat{q}_0| = 3$. If $\hat{q}_0 \neq q_0$ Lemma 3 implies $|\hat{q}_0| > |\bar{q}| + 1$. Consequently, $\hat{q}_0 = q_0$. Then $|q_0| = 3$ and $|q| = 8$, and Equation (6) yields $\sqrt[*]{P_q} \subseteq \{q_0, v, q\}$ where $v \sqsubset q$ and $|v| = |q| - 1 = 7$. Thus $p_q(t) \geq t^8 - t^5 - t - 1 = p_8(t) - t^2 \cdot p_3(t)$ for $1 \leq t \leq \lambda_3$.

This shows $p_q(\lambda_8) \geq -\lambda_8^2 \cdot p_3(\lambda_8) > 0$, that is, $\lambda_q < \lambda_8$. □

Summarising, the results of Section 8 yield

**Theorem 7.** *If $q \in X^*, |q| \geq 3$, is a reducible quasiperiod then $\lambda_q < \lambda_{|q|}$.*

Our main theorem (Theorem 3) then follows from Theorems 5 and 7.

Together with Corollary 12 our theorem yields a new proof of a theorem of [5] which shows that multi-scale quasiperiodic infinite words have zero topological entropy. In [5] a *multi-scale quasiperiodic infinite word* is a quasiperiodic infinite word which admits infinitely many quasiperiods.

## 9. Concluding Remark

In this paper we dealt with the function $f(\xi, n) = |\mathbf{infix}(\xi) \cap X^n|$ for quasiperiodic $\omega$-words. Their factor complexity (or topological entropy) is defined as $\tau(\xi) := \lim_{n \to \infty} \frac{\log_{|X|} |\mathbf{infix}(\xi) \cap X^n|}{n}$ (e.g., [4], Section 4.2.2 or [5,22]). Thus the upper bound for $\xi \in P_q^\omega$ is $\log_{|X|} \lambda_q \leq \log_{|X|} t_P$ which is bounded away from the value 1 for almost periodic $\omega$-words.

Along with the subword complexity in [5] the Kolmogorov complexity of quasiperiodic $\omega$-words was addressed. Obviously, subword complexity upper bounds Kolmogorov complexity (e.g., [22]). Since the $\omega$-languages $P_q^\omega$ are regular ones, the results of [22] show that there are $\omega$-words $\xi \in P_q^\omega$ whose Kolmogorov complexity achieves their subword complexity. Moreover, as $P_q^\omega = q \cdot R_q^\omega$ where $R_q^\omega$ is a finite prefix code, the results of [22,26,27] give more detailed bounds for most complex quasiperiodic $\omega$-words w.r.t. several notions of Kolmogorov complexity [28].

**Funding:** This research received no external funding.

**Institutional Review Board Statement:** Not applicable.

**Informed Consent Statement:** Not applicable.

**Data Availability Statement:** Not applicable.

**Conflicts of Interest:** The author declares no conflict of interest.

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
