# Peer review of "The Maximal Complexity of Quasiperiodic Infinite Words"

_axioms, doi:10.3390/axioms10040306_

Round 1

Reviewer 1 Report

Review of the Manuscript:

The maximal complexity of quasiperiodic infinite words

The manuscript outlines results about subword complexity. Presents a characterization of the set of infinite strings having a word q as quasiperiod. The characterization is used to calculate its maximal subword complexity and to characterize the quasiperiod of maximally complex quasiperiodic infinite strings. 
The problem under investigation is current, and the results answer important questions raised as a consequence of Solomon Marcus's works.

Being a special issue about Solomon Marcus, the author assume specialized knowledge from the reader. For this type of reader, the paper is easy to follow, the proofs are clear, and in my opinion, presents a unified view of the subject.

In my copy of the manuscript, the reference section appears twice.

Author Response

I run the manuscript through a spell checking procedure. Some typos are corrected.

Reviewer 2 Report

The paper deals with the quasiperiodic infinite words. After characterizing such words in terms of some finite language, the author mainly considers the subword/factor complexity of such words. In particular, Theorem 3 shows that words of the form a^nba^n or a^nbba^n generate the most complex infinite strings having the aforementioned words as quasiperiod. The paper contains some meaningful results, noteworthy Theorem 3. The techniques involved are non-trivial and some proofs are quite technical involving several cases, and they appear to be sound. The paper is nicely written and I have very few remarks and suggestions:

  • page 3 at line 14: there is an extra dot (..);
  • page 3 at line -11: prefixes of strings in->prefixed of THE strings in;
  • page 4 at line -14: provided for-> provided THAT for
  • Proposition 2 at page 6, in the third property: since you use a dual argument I would expect the suffix of q, not pref(q).
  • In Definition 2 at page 10: it is missing an IF in “if and only”. I would write: “if and only if in Q_p it is maximal w.r.t…”
  • The reference is written twice.

Author Response

I run the manuscript through a spell checker, corrected the mentioned typos and some others.

  • Proposition 2 at page 6, in the third property: since you use a dual argument I would expect the suffix of q, not pref(q).

I mentioned that the third item follows from the second one---thus pref is correct.

  • In Definition 2 at page 10: it is missing an IF in “if and only”. I would write: “if and only if in Q_p it is maximal w.r.t…”

I replaced "in" by "if", so the sentece sounds

“if and only if Q_p is maximal w.r.t…”

Round 2

Reviewer 1 Report

In my opinion, the manuscript is ready for publication.